# Impulse Control Disorders in the Polish Population of Patients with Parkinson’s Disease

**DOI:** 10.3390/medicina59081468

**Published:** 2023-08-16

**Authors:** Mateusz Toś, Anna Grażyńska, Sofija Antoniuk, Joanna Siuda

**Affiliations:** 1Department of Neurology, Faculty of Medical Sciences in Katowice, Medical University of Silesia, 40-055 Katowice, Poland; mateusz.tos@sum.edu.pl; 2Department of Imaging Diagnostics and Interventional Radiology, Kornel Gibiński Independent Public Central Clinical Hospital, Medical University of Silesia, 40-055 Katowice, Poland; grazynskaanna@gmail.com; 3St. Barbara Regional Specialist Hospital No. 5, 41-200 Sosnowiec, Poland; sofija.antoniuk@gmail.com

**Keywords:** impulse control disorders, Parkinson disease, dopamine agonists, compulsive buying, hypersexuality, pathological gambling, binge eating

## Abstract

*Background and Objectives:* Parkinson’s disease (PD) is one of the most common neurodegenerative diseases in the world. It is characterized by the presence of not only typical motor symptoms but also several less known and aware non-motor symptoms (NMS). The group of disorders included in the NMS is Impulse Control Disorders (ICDs). ICDs are a group of disorders in which patients are unable to resist temptations and feel a strong, pressing desire for specific activities such as gambling, hypersexuality, binge eating, and compulsive buying. The occurrence of ICDs is believed to be associated primarily with dopaminergic treatment, with the use of dopamine agonists (DA), and to a lesser extent with high doses of L-dopa. The aim of our study was to develop a profile of Polish ICDs patients and assess the frequency of occurrence of ICDs, as well as determine the risk factors associated with these disorders against the background of the PD population from other countries. *Materials and Methods:* Our prospective study included 135 patients with idiopathic PD who were hospitalized between 2020 and 2022 at the Neurological Department of University Central Hospital in Katowice. In the assessment of ICDs, we used the Questionnaire for Impulsive-Compulsive Disorders in Parkinson’s Disease (QUIP). Other scales with which we assessed patients with PD were as follows: MDS-UPDRS part III and modified Hoehn–Yahr staging. Clinical data on age, gender, disease duration and onset, motor complications, and medications were collected from electronic records. *Results:* ICDs were detected in 27.41% of PD patients (binge eating in 12.59%, hypersexuality in 11.11%, compulsive buying in 10.37%, and pathological gambling occurred in only 5.19% of patients. In total, 8.89% had two or more ICDs). The major finding was that ICDs were more common in patients taking DA than in those who did not use medication from this group (83.78% vs. 54.07%, respectively; *p* = 0.0015). Patients with ICDs had longer disease duration, the presence of motor complications, and sleep disorders. An important finding was also a very low detection of ICDs in a routine medical examination; only 13.51% of all patients with ICDs had a positive medical history of this disorder. *Conclusions:* ICDs are relatively common in the population of Polish PD patients. The risk factors for developing ICDs include longer duration of the disease, presence of motor complications, sleep disorders, and use of DA and L-dopa. Due to the low detectability of ICDs in routine medical history, it is essential for physicians to pay more attention to the possibility of the occurrence of these symptoms, especially in patients with several risk factors. Further prospective studies on a larger group of PD patients are needed to establish a full profile of Polish PD patients with ICDs.

## 1. Introduction

Idiopathic Parkinson’s Disease (PD) is the second most common neurodegenerative disease in the world. Aside from the typical motor disorders such as rigidity, tremor, and bradykinesia, patients are often accompanied by non-motor symptoms (NMS) such as cognitive, cardiovascular, autonomic, sleep, and psychiatric disorders. These disorders have a significant influence on the quality of life and overall functioning of the patients; they burden their interpersonal relationships and increase the chances of the eventual burnout of their caretakers [1,2,3].

The group of non-motor disorders occurring in PD patients also includes impulse control disorders (ICDs). ICDs are a group of psychiatric disorders that are characterized by the inability to resist temptations, impulses, or desires, despite their potential harm to the patient and their environment. ICD patients compulsively make decisions based on reward and gratification systems, without regard to the potential consequences [4,5]. These disorders are usually egosyntonic and thus are described by the patient as accepted and not requiring changes. However, with the progression of PD, motivation may change and be less oriented toward seeking pleasure and more towards relief of stress and tension [6]. The most commonly occurring ICDs are hypersexuality, compulsive buying, pathological gambling, and binge eating [7]. Additionally, ICD-related behaviors (ICD-RB) can be distinguished: dopamine dysregulation syndrome (DDS) involving usage of higher than recommended doses of dopaminergic drugs, punding, which are compulsive repetitions of useless activities such as folding and unfolding objects, hobbyism (excessive interest in particular activities), pathological hoarding and intensified, aimless walkabouts [8].

It is believed that ICDs occurrence is mostly related to dopaminergic treatment, especially the use of dopamine agonists (DA) and less to high doses of L-dopa [9]. Some studies show ICDs occurring with monoamine oxidase-B inhibitors, amantadine, and deep brain stimulation (DBS) treatments [10,11,12]. The incidence of ICDs in PD is vastly different depending on the population, and can be from 23.48% in the Spanish population up to 34.8% in the Finnish population. These discrepancies can result from differences in diagnostic tools, diagnostic criteria applicable in different parts of the world, and underdiagnosing by clinicians [13,14]. Self-reporting of the symptoms by the patients also needs to be taken into account where cultural differences and the embarrassing nature of some symptoms may have a substantial impact on informing doctors of the course of the disease, and have a significant impact on the final diagnosis.

Our observations show a lack of publications assessing ICDs occurrence in the Polish population of PD patients. The aim of our study was to develop a profile of Polish ICDs patients and assess the frequency of occurrence of ICDs, and their subtypes, as well as compare clinical differences between ICDs and non-ICD patients and determine the sociodemographic and clinical risk factors associated with these disorders against the background of the PD population from other countries.

## 2. Materials and Methods

### 2.1. Subjects and Data Collection

This study was prospective and involved 135 Polish patients with idiopathic Parkinson’s disease (PD) admitted to the Neurological Department of University Central Hospital in Katowice for regular control visits or therapy modification. During hospitalization, patients were provided with information about this study and were offered to participate. The duration of the prospective data collection phase was from November 2020 to September 2022. The inclusion criteria were: (a) confirmed diagnosis of PD, according to the Movement Disorder Society Clinical Diagnostic Criteria for Parkinson’s disease [15], (b) the patient’s condition enabling the completion of the questionnaire. The exclusion criteria were (a) moderate or severe dementia, (b) neurodegenerative diseases other than PD (c) severe motor conditions preventing completion of the questionnaire. Participation in this study was proposed to 236 patients, of whom 163 agreed to participate in the study; however, 28 patients were excluded from the study due to the inclusion and exclusion criteria. Due to the prospective nature of this study, the local ethics committee of the Medical University of Silesia approved the study (decision number PCN/0022/KB1/99/I/19/21). Written consent of the participants for their participation in this study was obtained. All the test procedures were carried out in compliance with the ethical principles of the 1964 Helsinki Declaration and its subsequent amendments.

Demographic data (including age, sex, and smoking) and primary medical data were collected from medical history as well as patient history (including information on the occurrence of first symptoms, disease duration, treatment, motor complications, and comorbidities).

### 2.2. Study Procedure and Instruments Used

In the first part of this study, the data on ICDs incidence were taken from routine medical history collection; then, it was evaluated using the Polish translation of the Questionnaire for Impulsive Compulsive Disorders in Parkinson’s Disease (QUIP) licensed from the University of Pennsylvania [16]. QUIP is a self-completed questionnaire based on DSM-IV classifications. QUIP is one of the most commonly used and recommended questionnaires for ICD screening [17]. QUIP has very good clinimetric properties, enabling the detection of ICDs with greater sensitivity and specificity than in the case of other scales. It comprises three sections, including questions on ICDs and the most prevalent ICD-RBs (with introductory descriptions of the disorder). Cut-off points for specific ICDs were used as the authors of the original validation paper proposed as well as other authors used: (a) pathological gambling: ≥2 positive answers, (b) hypersexuality: ≥1, (c) compulsive shopping: ≥1, and (d) binge eating: ≥2 answers [18,19,20].

Disease severity was assessed by neurologists with experience in neurodegenerative diseases with Polish validation of MDS-UPDRS part III [21] and modified Hoehn and Yahr staging both in the “OFF” and “ON” states [22]. Cognitive disorders were assessed by a trained psychologist using Mini-Mental State Examination (MMSE) [23] and Addenbrooke’s Cognitive Examination-III (ACE-III) [24], whereas depression was assessed using Polish adaptation of the Beck Depression Inventory (BDI-II) [25]. A movement disorders specialist assessing motor symptoms was blinded to the QUIP score assessed by another physician, who, at the time of the assessment, was blinded to the scores of motor symptoms scales. The authors have permission to use this instrument from the copyright holders. The assessment of the influence of dopaminergic treatment on the incidence of ICDs was conducted by separately counting the total levodopa equivalent daily dose (LEDD), LEDD for levodopa alone, and LEDD for dopamine agonists only [26]. The calculation allowed us to assess the effect of the size of both the total dose of dopaminergic drugs as well as the person’s DA and L-dopa individually. Both groups of drugs are considered the primary risk factors for the development of ICDs.

### 2.3. Statistical Analysis

To determine the size of the study group, an a priori power analysis was performed using G*Power 3.1. A total sample size of 128 was obtained, assuming a medium effect size and power set to 0.80. Statistica 13.0 software (TIBCO Software Inc., Palo Alto, CA, USA) was used for all of the statistical analyses. Data were reported as mean ± standard deviation (SD) for continuous data and n (%) for categorical data. The Shapiro–Wilk test was applied to assess the normality of the quantitative variables. Student’s *t*-test was used to compare the continuous variables with normal distribution, the Mann–Whitney U-test was used to compare continuous variables with non-normal distribution, and the chi-square test for categorical variables to compare differences between patients with and without ICDs. The level of statistical significance was determined for *p* < 0.05.

## 3. Results

Out of 163 patients taking part in this study, six were excluded due to negative verification of diagnosis of idiopathic Parkinson’s Disease, four due to cognitive disorders that made filling out the questionnaire impossible, and 18 patients due to incomplete QUIP questionnaires. Finally, 135 PD patients were qualified for this study; their clinical profiles are shown in Table 1. The most commonly used treatment was the combination of L-dopa with DA—57.04% of patients. L-dopa alone was used in 31.85% of patients (89.63% in total, taking into account independent use and in combination with DA). DA in monotherapy was used by 5.19% of patients (62.22% in total, taking into account independent use and in combination with L-dopa). Among the DA, the most commonly used was ropinirole—47.41% of patients, then pramipexol—10.37%, then piribedil—2.22% and rotigotine—0.74%, whereas continuous subcutaneous apomorphine injections were used by 1.48% of respondents. Less commonly used anti-parkinsonian drugs were amantadine in 31.85% of patients, MAO-B inhibitors in 20.74% of patients, and anticholinergic drugs in 7.41%. An additional ten patients (7.41%) were on subthalamic nucleus deep brain stimulation (DBS-STN) treatment, and 2 (1.48%) were treated with continuous intestinal infusion of L-dopa/carbidopa gel.

### 3.1. Frequency of ICDs

The overall incidence of ICD was 27.41% (37 out of 135 patients). The most prevalent ICD was binge eating in 17 (12.59%) patients, subsequently hypersexuality in 15 patients (11.11%), compulsive buying in 14 (10.37%), and pathological gambling occurred in only (5.19%) 7 patients. Criteria for more than one ICD were met by 12 patients (8.89%).

During a routine doctor’s evaluation, only five patients (3.70%) had shown a history of ICDs, which constituted 13.51% of all patients meeting the ICDs criteria using QUIP evaluation. According to the QUIP questionnaire, any ICD-RB was positive for 36 (26.67%) patients (including hobbyism in 20, punding in 27, and walkabout in 8).

### 3.2. Correlates of ICDs

Comparing ICDs patients with non-ICDs, there were no significant differences in sex, age, smoking frequency, cognitive status, and age of diagnosis of the disease. Patients did not differ in terms of disease severity expressed in H&Y and MDS-UPDRS part III scales. However, it had been observed that ICDs patients, compared to non-ICDs, have longer disease duration (11.86 ± 4.87 vs. 9.97 ± 6.35, respectively; *p* = 0.0341), more frequently they had motor complications such as levodopa-induced dyskinesia and wearing-off phenomenon (83.78% vs. 58.16%, respectively; *p* = 0.0053) and sleep disorders (67.56% vs. 47.96% accordingly; *p* = 0.0417). Particular types of ICDs and ICD-RBs, depending on sex, were: hypersexuality and hobbyism were more common in men (15.73% vs. 2.17%; *p* = 0.0175 and 19.10% vs. 6.52%; *p* = 0.0387, respectively) (Figure 1).

As expected, ICDs were more common in patients taking DA than in those who did not use medication from this group (83.78% vs. 54.07%, respectively; *p* = 0.0015) (Table 2). No statistically significant differences in ICD were seen between patients using ropinirole and pramipexole (*p* = 0.2921). As for other DA: interestingly, patients receiving apomorphine in continuous subcutaneous infusion and rotigotine in transdermal patch did not show any ICD symptoms, whereas, in patients that received piribedil, the criteria for more than one ICD was met by one patient. ICDs were present in 37.67% of patients receiving combined treatment of DA and L-dopa, 28.58% in patients using DA without L-dopa, and 13.3% in patients using L-dopa without DA (*p* = 0.0127). No influence on ICD occurrence has been shown for other anti-parkinsonian drugs, nor DBS-STN. There was no ICD occurrence in patients using either L-dopa or DA.

## 4. Discussion

To the best of our knowledge, the present study is the first based on Polish patients with idiopathic Parkinson’s disease and one of few studies containing populations of patients from central-eastern Europe. Notably, the Polish population is distinguished by the unique genetic landscape, which may have implications for the occurrence and manifestation of PD-related complications, including impulse control disorders [27]. In addition, the treatment algorithms used in our country differ from Western treatment regimens in that patients later receive DA, instead as an addition to L-dopa treatment, then as an independent therapy. We acknowledge that Poland is an ethnically and culturally distinct, traditionalist country. This societal backdrop may influence how mental health problems and neurological diseases, including ICDs, are perceived and discussed in public spaces. By studying ICD occurrence in the Polish population, we can shed light on the potential influence of sociocultural factors on the recognition, reporting, and treatment-seeking behavior of ICDs. Understanding these influences is essential for devising targeted and culturally sensitive interventions for PD patients with ICDs in Poland. Another crucial aspect is the need to thoroughly assess the prevalence of awareness of impulse control disorders in the Polish population. Lack of awareness about these disorders may lead to underdiagnosis and undertreatment, further exacerbating the negative impact on the patient’s quality of life. Conducting a comprehensive study on ICDs in PD patients can help raise awareness among healthcare professionals, patients, and their families, leading to early identification and appropriate management. Additionally, our research can be a foundation for developing educational initiatives to disseminate information about ICDs, their risk factors, and treatment options within the Polish population. In a group of 135 PD patients, 37, which constitutes 27.41%, were diagnosed with at least one type of ICD. The ICDs patient group was diverse in terms of age, sex, and disease span. The frequency of ICD occurrence in our analysis is comparable with other European research: for the ICARUS [28] study on the Italian population, it was 28.6% at the beginning of the study, 29.3% after a year, and 26.5% after two years of follow up. Comparable results were also seen from Spanish and French authors, where 23.48% and 25% of patients had ICDs, respectively [14,29]. A different result has been reached by other Spanish researchers [10,30], who concluded a significantly higher prevalence of ICDs than in our study, 58.3% for Vela et al. and 39.1% for Garcia-Ruiz et al. In Vela et al. study, the disproportions came from the fact that their study was based on the early onset PD population, whereas our study included the whole range of PD patients, disregarding the age of onset. Patients with early onset PD are more likely to experience more frequent and more severe psychiatric and behavioral problems such as depression and obsessive-compulsive disorders, with prevalence of these varying from 5 to 45% [31,32]. It is worth pointing out that in this relatively young group of patients, the most commonly occurring disorder was hobbyism (20% of the population), which may be connected to younger people’s easy access to tools such as smartphones, internet web pages, and internet games, which are linked with high hobbyism prevalence and can alter the results. It should also be noted that hobbyism was the most prevalent ICD in the control population. For Garcia-Ruiz et al., all patients included in the study were treated with DA, whereas for our patients, it was only 62%, which could have a significant impact on the overestimation of ICD patients in the Spanish research. Across the European population, significantly lower results were obtained for the Italian population [33], where ICDs were prevalent in only 8.1% of PD patients, which could be caused by the fact that authors of the paper also included patients with dementia in the study which constituted for 26.3% of the whole population, whereas patients with severe cognitive dysfunctions were excluded from our analysis and the analyses of other authors, due to the fact that in this cognitive state, the answers to QUIP questionnaire might have been inadequate, or the test could not be performed at all.

In the case of patients in Slavic countries or otherwise similar ethnically or culturally to the Polish population, the incidence of ICD in Marković et al. study [34] for the Serbian population was 19.8% at the beginning of the study and 29.2% after the 5-year follow-up, which makes these results comparable to ours. In the case of the Czech population [35] ICDs occurred in 26.5% of PD patients, which is comparable to our study; however, it should be noted that only early-onset PD patients were qualified for this study and a different scale measuring ICD occurrence was used—the South Oaks Gambling Screen and Modified Minnesota Impulse Disorders Interview. The disproportion of ICD prevalence in early-onset PD patients between our study and Gescheidt et al. and Vela et al. is most probably caused by different diagnostic tools used for ICD detection.

The most prevalent ICDs in our study were: binge eating which occurred in 12.59% of patients, hypersexuality (11.11%), and compulsive buying (10.37%). Our results do not differ from the European and American populations for which binge eating and compulsive buying were interchangeably the most common ICDs [36]. Pathological hazard was relatively rare—only 5.19% of patients, which is not in line with the results of other studies, where the pathological hazard is at the top of most common ICDs [37,38,39]. This may be due to the lesser popularity of this type of activity in our and other European countries, and it is much more prominent in Asian countries [40].

In our PD patients group, a few significant differences concerning demographic factors in ICD occurrence were observed. Most importantly, we have shown that ICD patients have been suffering from PD longer than non-ICD patients, which is consistent with literature data [41]. It is presumably a result of polytherapy, with most commonly DA added to L-dopa in this group. In contrast to previous research [42], no correlation between the younger age of patients and earlier PD onset on ICDs occurrence had been shown here, which may be due to the relatively low percentage of early onset PD enrolled in our study. There was also no correlation between smoking and ICDs incidence, which is consistent with some previously published research. As opposed to the ICARUS study [28] and according to the DOMINION study [43] and other Slavic research [35], no correlation was seen between gender and ICD prevalence. However, according to previous publications on this topic [44,45], we have concluded that in men, hypersexuality and hobbyism were more common than in women. The gender difference in the incidence of hypersexuality is possibly due to higher intrinsic sexual motivation in men, as well as greater ease in arousal and sociocultural conditioning [46]. We have not shown exacerbation of PD symptoms expressed with MDS-UPDRS part III and their severity with modified Hoehn-Yahr staging on ICD incidence, which supports the theory that Parkinson’s disease by itself does not lead to these types of disorders, which is in accordance with the previous research [47]. However, it should be noted that in our study, patients with ICDs were more likely to have motor complications such as levodopa-induced dyskinesia (LID) and wearing-off phenomenon distinctive for advanced disease. This association is most likely due to the fact that ICD patients experiencing motor fluctuations have longer disease duration than non-ICD patients and, thus, have a higher probability of longer exposure to DA, and they often use higher doses of other anti-parkinsonian drugs [48]. Moreover, it is believed that ICDs and LID share epigenetic and genetic mechanisms such as Ser9Gly gene polymorphism that codes D3 receptor, or higher expression of transcriptional regulator ΔFosB [49,50]. We have found out that ICD patients are more likely to have sleep disorders (including insomnia, hypersomnia, and REM sleep behavior disorder) compared to non-ICD patients, which is in accordance with previous analyses [51,52]; however, the mechanism of this correlation remains unknown and requires further research [53]. In our study group, no correlation between the intensity of depression and ICDs occurrence was shown, which is not consistent with results obtained by other researchers [54]. The lack of this correlation may be partially explained by the lower intensity of depressive symptoms in patients receiving higher doses of DA [55], being a well-known factor in the development of ICDs.

According to our assumptions and previous publications [56], ICD symptoms occurred significantly more often in patients receiving DA, and patients with ICDs have received higher doses of DA than non-ICD patients. No statistically significant differences in the prevalence of ICDs between patients receiving ropinirole and pramipexole were observed, which is also in accordance with previous publications [41] and suggests non-ergot DA to be a class of drugs affecting the development of ICDs [43]. Although the influence of second-generation DA on the development of ICDs has been documented in numerous studies, the mechanism of its ICD influence is not clear. Non-ergot DA, such as ropinirole and pramipexole, have a higher affinity to dopamine receptors D3 than D1 and D2, which is associated with their impact outside of the nigrostriatal pathway [57]. Dopamine replacement therapy (DRT) restores the normal levels of dopamine in motor pathways but can, at the same time, stimulate relatively conserved mesocortical pathways, which, for genetically susceptible patients, can lead to hypersensitivity of the reward system and stimulate ICD development [58]. Regarding the remaining DA—due to the very small group of patients that used them, we are unable to clearly determine their impact on ICDs incidence. Nevertheless, it should be noted that in our group of patients using transdermal or continuous subcutaneous infusion forms of DA, ICDs were practically non-existent. Some publications suggest that due to distinct pharmacokinetics, the transdermal form of rotigotine and the continuous subcutaneous infusion of apomorphine enable constant stimulation of dopamine receptors without excessive pulsatile stimulation of the receptors, which may potentially reduce the risk of developing ICDs [59,60]. The most commonly occurring ICDs were determined in patients using a combination of DA and L-dopa, which may suggest their role in ICDs development. Moreover, in the group using L-dopa monotherapy, we observed ICD patients to be using higher doses than non-ICD patients, which is consistent with other publications [61]. Contrary to a portion of previous studies and meta-analyses [9,62], we did not find any effect of amantadine on the incidence of ICDs, and similar results were obtained by some researchers [63,64]. Some authors believe that the inconclusive results on the effect of amantadine on ICD development may stem from its multidirectional activity, where the dopaminergic response may contribute to the development of ICDs and anti-glutaminergic to potentially reducing the risk of ICDs [65]. Finally, we did not determine any effect of DBS-STN system implantation on ICDs development, which is consistent with some previous studies [43,66]. However, in our study group, only 7.41% of patients had an implanted DBS-STN system, which makes it difficult to draw firm conclusions. The impact of DBS-STN on the development of ICDs is complex; on the one hand, implantation of the DBS system allows the reduction of doses of dopaminergic drugs, but on the other hand, it is believed that DBS-STN may lead to impairment of the pattern of global response inhibition, which in turn may lead to increased impulsivity and thus lead to development of ICDs [67].

Our study had some limitations. Firstly, it was a single-center study conducted on a relatively small group of patients with PD and, thus, a small group of patients with ICDs. Moreover, the study group came from only one region of the country, mainly the southern part of Poland. Another limitation is the fact that our main ICDs evaluation tool was the QUIP questionnaire, which is a satisfactory screening tool for detecting the existence of a given group of disorders, but does not allow differentiation between distinct subgroups of disorders, their severity, and their impact on a patient’s life. In everyday clinical practice, attention should be paid not only to the detection of a given disorder, but also to undertaking a further diagnostic process in order to better understand the complexity of a problem, whether a given disorder is definitely due to PD or other factors faced by the patient, and to select a procedure that would improve the patient’s quality of life as much as possible.

However, the role that screening tools such as QUIP play in drawing attention to the problem of ICDs by both clinicians and patients themselves cannot be underestimated. The results of our study showed that in a routine neurological evaluation, only 3.7% of patients and physicians noted the occurrence of disturbing symptoms from the spectrum of impulse control disorders, and after screening with the use of the questionnaire, this percentage increased to 26.67%. It means that patients are not aware of their symptoms in the form of ICDs, their relation to PD, and their ability to pharmacologically control them. Doctors often do not evaluate these symptoms, not considering them as essential to patient wellbeing, or forget to ask about them. Therefore, we recommend that ICD forms such as QUIP should be routinely used in patients with PD, particularly using DA, in order to improve their health-related quality of life.

## 5. Conclusions

ICDs are relatively common in the population of Polish PD patients. The risk factors for developing ICDs include longer duration of the disease, presence of motor complications, sleep disorders, and use of DA and L-dopa. The population of Polish patients with ICDs in our study was comparable to other European populations, considering similar patient demographics, time of onset of the disease, and the treatment used. As with other European and American populations, the most common ICDs were binge eating, hypersexuality, and compulsive buying, which is different from, for example, Asian populations where gambling often predominates. As in other Slavic populations, there was no difference between genders in the prevalence of ICDs. However, we showed a significant advantage in most hypersexuality and hobbyism in men. Due to the low detectability of ICDs in routine medical history, it is essential for physicians to pay more attention to the possibility of the occurrence of these symptoms, especially in patients with several risk factors. Early ICDs diagnosis is crucial in their effective therapy; their awareness reduces the sense of shame in patients and improves the quality of life of patients and their caregivers. Further prospective studies on a larger group of PD patients are needed to establish a full profile of Polish PD patients with ICDs.

## Figures and Tables

**Figure 1 medicina-59-01468-f001:**
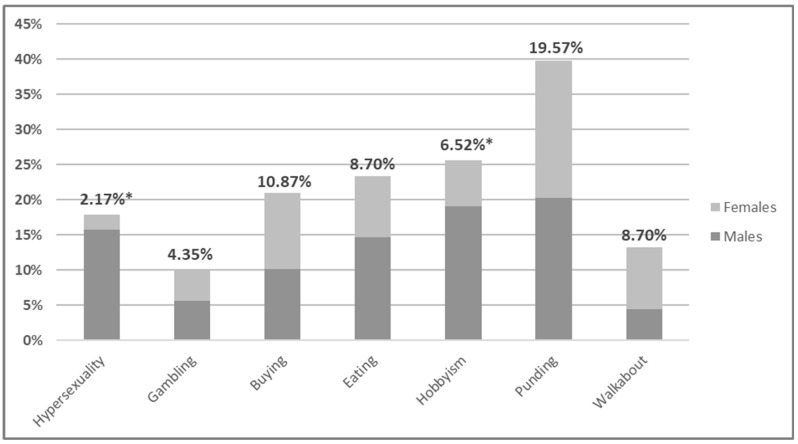
Gender distribution and frequency across ICDs and ICD-RBs. ICDs = Impulse control disorders; ICD-RBs = Impulse control disorder-related behaviors; * significant difference at *p* < 0.05 in distribution between genders.

**Table 1 medicina-59-01468-t001:** Clinical features of the study population.

N: 135	DA Type:
Gender: male 89 (65.93%); female 46 (34.07%)	Ropinirole: 64 (47.41%)
Age: 63.24 ± 9.56 years	Pramipexole: 14 (10.37%)
Smoking: 6 (4.44%)	Piribedil: 3 (2.22%)
MDS-UPDRS part III in “OFF state”: 40.15 ± 17.37	Rotigotine: 1 (0.74%)
MDS-UPDRS part III in “ON state”: 18.81 ± 11.77	DA-LEDD (mg): 144.96 ± 211.32
Age of onset: 52.87 ± 10.91	MAO-B inhibitors: 28 (20.74%)
Disease Duration: 10.49 ± 6.02	Amantadine: 43 (31.85%)
Motor fluctuation: 88 (65.19%)	Levodopa: 121 (89.63%)
Sleep disorders: 72 (53.33%)	LD-LEDD (mg): 845.81 ± 565.49
	Total LEDD (mg): 1095.52 ± 565.49

Data are shown as numbers and percentages for qualitative variables and mean ± SD for quantitative variables. DA—dopamine agonist; LEDD—levodopa equivalent daily dose; LD—levodopa; MDS-UPDRS—MDS Unified Parkinson’s Disease Rating Scale.

**Table 2 medicina-59-01468-t002:** Association of clinical features and ICD.

	ICDs Patients	Non-ICDs Patients	*p*
Male	70.27%	64.29%	0.5127
Age (years)	63.57 ± 8.97	63.12 ± 9.80	0.8319
Age of onset (years)	51.89 ± 9.59	53.24 ± 11.39	0.5256
Disease duration (years)	11.86 ± 4.87	9.97 ± 6.35	0.0341
MDS-UPDRS part III in “OFF state”	38.28 ± 16.29	40.84 ± 17.78	0.4931
MDS-UPDRS part III in “ON state”	18.37 ±12.29	18.97 ± 11.64	0.6499
Motor fluctuation	83.78%	58.16%	0.0053
Sleep disorders	67.56%	47.96%	0.0417
Dopamine agonist use	83.78%	54.08%	0.0015
DA-LEDD	170.78 ± 120.15	135.25 ± 236.60	0.0217
MAO-B inhibitors	27.03%	18.37%	0.2683
Amantadine	35.14%	30.61%	0.6149
Levodopa	94.59%	87.76%	0.2449
LD-LEDD	804.68 ± 536.98	954.73 ± 629.73	0.3038
LD-LEDD (patients receiving only monotherapy)	1625 ± 944.6	916.90 ± 527.20	0.0489
Total LEDD	1268.24 ± 645.56	1030.31 ± 624.33	0.0524

Data are shown as numbers and percentages for qualitative variables and mean ± SD for quantitative variables. DA—dopamine agonist; LEDD—levodopa equivalent daily dose; LD—levodopa; MDS-UPDRS—MDS Unified Parkinson’s Disease Rating Scale.

## Data Availability

The datasets generated and analyzed during the current study are available on request from the corresponding author.

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
