# Peer review of "Impulse Control Disorders in the Polish Population of Patients with Parkinson’s Disease"

_medicina, 2023, doi:10.3390/medicina59081468_

Round 1

Reviewer 1 Report

In the manuscript entitled “Impulse Control Disorders in the Polish population of patients with Parkinson's disease” the authors investigated the Impulse control disorders among patients with Parkinson's disease receiving dopamine in Polish population. These studies are especially helpful in studying neurobiology of impulsive forms of behavior (such as problem gambling) that appear to be caused, in part, by the therapeutic use of dopamine receptor agonists. In my opinion the article is well written, and I trust that the manuscript will be of interest to potential readers.  

Author Response

Dear reviewer,

Thank you very much for your kind attitude and positive assessment of our work.

Reviewer 2 Report

Dear authors, it is a pleasure to review your manuscript. I would like to make a number of improvements to the text:

-The signature of the informed consent form is not considered an inclusion criterion, but a requirement to take part in the study. The lack of a signed informed consent form should also not be included in the exclusion criteria.

-What is the reliability and validity of the questionnaires used?

-How were the neurologists and psychologists who participated in the study recruited? Did they receive any information/training regarding the study prior to its initiation? What consideration was offered to them in exchange for participating in the study?

Author Response

Dear Reviewer,

Thank you for your valuable and constructive comments and suggestions that will make our article better. Below are point-by-point answers to your evaluation.

“The signature of the informed consent form is not considered an inclusion criterion, but a requirement to take part in the study. The lack of a signed informed consent form should also not be included in the exclusion criteria.”

We have removed informed consent information from the inclusion and exclusion criteria

“-What is the reliability and validity of the questionnaires used? “

According to the authors of QUIP, the scale is characterized by high validity and sensitivity (validity for each disorder [AUC]: pathological gambling = 0.95, hypersexuality = 0.97, compulsive buying = 0.87 and compulsive eating = 0.88; the sensitivity of the QUIP to detect any disorder was 96%).

We have added short information about this in the text of the manuscript.

“How were the neurologists and psychologists who participated in the study recruited? Did they receive any information/training regarding the study prior to its initiation? What consideration was offered to them in exchange for participating in the study?”

The neurologists who were performing the motor assessment of patients are employees of the hospital where the study was conducted as well as specialists in the field of movement disorders dealing with patients with PD on a daily basis. In addition, a significant part of the patients was personally assessed by the authors of the study: Joanna Siuda and Mateusz ToÅ›. The Movement disorders specialist assessing motor symptoms was blinded to the QUIP score assessed by another physician, who at the time of the assessment was blinded to the scores of motor symptoms scales. The psychologist is also an employee of the hospital where the study was conducted and she is specialized in the diagnosis of patients with neurodegenerative diseases and has the appropriate certification. All specialists involved in the study were informed about the purpose of the study. In addition, the procedures they performed, i.e. neurological and psychological examination, are routine in our center. All additional procedures, i.e. collecting demographic data, calculating LEDD, and performing QUIP, were conducted by the author and co-authors of the study. No one received any additional remuneration for participating in this project.

We have added short information about the blinding of researchers in the section Materials and Methods of the manuscript.

Reviewer 3 Report

This is an interesting study examining impulse control disorders in the Polish population of patients with Parkinson's disease. The paper is well-written and I agree that it may contribute well to the literature. I have several comments to improve the manuscript further:

1. First, the authors highlight the lack of publications on ICD occurrence in the Polish population of PD patients. It might be beneficial to briefly discuss the potential importance of conducting such a study and how it could contribute to the existing knowledge on ICDs in Parkinson's disease.

2. The study is described as prospective, which is appropriate for collecting data over a specific period. However, it might be beneficial to further clarify the duration of the prospective data collection phase (i.e., from November 2020 to September 2022)

3. More citations should be used to support the cut-off points for specific ICDs used in the current study.

4. The methodology for assessing the influence of dopaminergic treatment on ICD incidence is well-described by counting the levodopa equivalent daily dose (LEDD) for different treatment components. However, There should be more explanation of LEDD and its significance in the context of the study.

5. There should be justification regarding the sample size used in the current study

6. The authors should also elaborate on the number of missing data and whether missing data imputation was used.

7. The authors compare their findings with previous studies from different European countries. It will be useful to discuss similarities and differences in ICD prevalence and types, which helps contextualize the results within the existing literature.

8. The sample size is quite small and should be acknowledged in the limitation section

Author Response

Dear Reviewer,

Thank you for your valuable and constructive comments and suggestions that will make our article better. Below are point-by-point answers to your evaluation.

”1. First, the authors highlight the lack of publications on ICD occurrence in the Polish population of PD patients. It might be beneficial to briefly discuss the potential importance of conducting such a study and how it could contribute to the existing knowledge on ICDs in Parkinson's disease.”

We added missing information to the discussion on lines 210-229 with the content: Notably, the Polish population is distinguished by the unique genetic landscape, which may have implications for the occurrence and manifestation of PD-related complications, including impulse control disorders. In addition, the treatment algorithms used in our country differ from Western treatment regimens in that patients later receive DA, instead as an addition to L-dopa treatment, then as an independent therapy. We acknowledge that Poland is an ethnically and culturally distinct, traditionalist country. This societal backdrop may influence how mental health problems and neurological diseases, including ICDs, are perceived and discussed in public spaces. By studying ICD occurrence in the Polish population, we can shed light on the potential influence of sociocultural factors on the recognition, reporting, and treatment-seeking behaviour of ICDs. Understanding these influences is essential for devising targeted and culturally sensitive interventions for PD patients with ICDs in Poland. Another crucial aspect is the need to thoroughly assess the prevalence of awareness of impulse control disorders in the Polish population. Lack of awareness about these disorders may lead to underdiagnosis and undertreatment, further exacerbating the negative impact on the patient's quality of life. Conducting a comprehensive study on ICDs in PD patients can help raise awareness among healthcare professionals, patients, and their families, leading to early identification and appropriate management. Additionally, our research can be a foundation for developing educational initiatives to disseminate information about ICDs, their risk factors, and treatment options within the Polish population.

”2. The study is described as prospective, which is appropriate for collecting data over a specific period. However, it might be beneficial to further clarify the duration of the prospective data collection phase (i.e., from November 2020 to September 2022).”

As suggested, we clarify in the manuscript that the duration of the prospective data collection phase was from November 2020 to September 2022.

”3. More citations should be used to support the cut-off points for specific ICDs used in the current study. ”

We added in the manuscript additional citations from the following studies by other authors who adopted the same cut-off point in the QUIP:

  • Maréchal E, Denoiseux B, Thys E, Cras P, Crosiers D. Impulsive-Compulsive Behaviours in Belgian-Flemish Parkinson's Disease Patients: A Questionnaire-Based Study. Parkinsons Dis. 2019 Mar 18;2019:7832487.
  • Sharma A, Goyal V, Behari M, Srivastva A, Shukla G, Vibha D. Impulse control disorders and related behaviours (ICD-RBs) in Parkinson's disease patients: Assessment using "Questionnaire for impulsive-compulsive disorders in Parkinson's disease" (QUIP). Ann Indian Acad Neurol. 2015 Jan-Mar;18(1):49-59.
  • Paul BS, Aggarwal S, Paul G, Khehra AS, Jain A. Impulse-Control Disorders and Restless Leg Syndrome in Parkinson's Disease: Association or Coexistence. Ann Indian Acad Neurol. 2023 Mar-Apr;26(2):161-166.

“4. The methodology for assessing the influence of dopaminergic treatment on ICD incidence is well-described by counting the levodopa equivalent daily dose (LEDD) for different treatment components. However, There should be more explanation of LEDD and its significance in the context of the study.”

In 2.2. Study procedure and instruments used section, we have added the following information on the significance of LEDD for different treatment components: The calculation allowed us to assess the effect of the size of both the total dose of dopaminergic drugs as well as the person's DA and L-dopa individually. Both groups of drugs are considered the primary risk factors for the development of ICDs.

“5. There should be justification regarding the sample size used in the current study”

We performed a priori power analysis using G*Power 3.1 obtaining a total sample size of 128 assuming medium effect size and power set to 0.80. We have added information about this in the manuscript in the Materials and Methods section.

“6. The authors should also elaborate on the number of missing data and whether missing data imputation was used.”

As the motor assessment of patients and the neuropsychological examination are routine procedures performed in our center, we did not have missing data in this regard. Patients who returned incompletely filled out the QUIP scale were disqualified from the study. For this reason, we did not need to use imputation of missing data.

“7. The authors compare their findings with previous studies from different European countries. It will be useful to discuss similarities and differences in ICD prevalence and types, which helps contextualize the results within the existing literature.”

In our manuscript's "conclusions" section, we have provided a comprehensive summary of our study population also comparing it with other European populations (lines 373-380).

“8. The sample size is quite small and should be acknowledged in the limitation section”

Information about the Small group of subjects as a limitation of the study is pointed out in the manuscript in paragraphs 348-349 and has the following text: Our study had some limitations. Firstly, it was a single-centre study conducted on a relatively small group of patients with PD and thus a small group of patients with ICDs.

Reviewer 4 Report

While the manuscript is well written there are some paragraph formatting issues that should be addressed.

Introduction:

Please break down the first paragraph into multiple paragraphs.

Methodology:

How were participants recruited? 

Was there an a priori power analysis?

How many total participants were originally recruited and how many agreed to participate?

Was the QUIP administered in Polish or English?

You should break down your methodology into sections (e.g. participants, instruments, procedure, statistical analysis section)

How far apart were each of the exams administered? How frequently did the subjects come to get assessed?

In your statistical analysis you stated that groups were compared, but nowhere in your objectives did you share with the readers that you were planning to compare any groups. Please modify your objectives in your introduction to match what was done (identified differences between patients with ICDs and no ICDs)

Results:

The results are well written

Discussion

Overall the discussion is well written however, another limitation in your study is that most of your participants were male.

I will withold on assessing the entire discussion section without more information that could change the discussion based on the methodology that needs to be expanded upon.

There were several paragraphs that need to be broken up. 

Author Response

Dear Reviewer,

Thank you for your valuable and constructive comments and suggestions that will make our article better. Below are point-by-point answers to your evaluation.

“Introduction: Please break down the first paragraph into multiple paragraphs.”

In the introduction, we divided the first paragraph into three separate ones.

“Methodology: How were participants recruited?”

Patients diagnosed with Parkinson's disease who were hospitalized in our center for a follow-up visit or modification of therapy were provided with information about the study and offered to participate in it. We added this information in the 2.1.1 Subjects and data collection subsection of the manuscript.

“Was there an a priori power analysis?”

We performed an a priori power analysis obtaining a total sample size of 128 assuming medium effect size and power set to 0.80

“How many total participants were originally recruited and how many agreed to participate?”

Participation in the study was proposed to 236 patients, among them 163 agreed to participate in the study.

“Was the QUIP administered in Polish or English?”

QUIP was administered in the Polish language version. We obtained a license to use the Polish translation of  QUIP based on an agreement with the University of Pennsylvania. We added information about the use of the Polish translation of QUIP in the manuscript in the Materials and Methods Instruments section.

“You should break down your methodology into sections (e.g. participants, instruments, procedure, statistical analysis section)”

We have divided the Materials and Methods section into subsections.

“How far apart were each of the exams administered? How frequently did the subjects come to get assessed?”

The assessments of patients were made once during a routine hospital and were not repeated. The MDS-UPDRS part. III evaluation was performed both in the “OFF” and “ON” states.

“In your statistical analysis, you stated that groups were compared, but nowhere in your objectives did you share with the readers that you were planning to compare any groups. Please modify your objectives in your introduction to match what was done (identified differences between patients with ICDs and no ICDs).”

We added information about the comparison of ICDs and non-ICDs patients in the study objectives.

“Overall the discussion is well written however, another limitation in your study is that most of your participants were male”

Thank you for appreciating our discussion. More male participants in our study are due to epidemiology which shows that PD is more common in men [1], therefore our group is representative of this patient population.

  1. Haaxma CA, Bloem BR, Borm GF, Oyen WJ, Leenders KL, Eshuis S, Booij J, Dluzen DE, Horstink MW. Gender differences in Parkinson's disease. J Neurol Neurosurg Psychiatry. 2007 Aug;78(8):819-24. doi: 10.1136/jnnp.2006.103788.

Round 2

Reviewer 3 Report

The authors have addressed all my comments well. I appreciate their efforts.

Author Response

Dear reviewer,

Thank you for your positive reception and acceptance of our amendments.

Reviewer 4 Report

I appreciate the authors thoroughness in responding to my concerns. 

The only edits left are the questions that the authors responded to in their responses to me but did not add to the manuscript. Please add your a priori power analysis, recruitment and such into your manuscript.

Author Response

Dear reviewer,

Thank you for accepting and appreciating our answers. As suggested, we have added missing information about priori power analysis (paragraphs 140-142), participant recruitment (paragraphs 97-99), and motor evaluation (paragraph 125) into the text of the manuscript.